# The Role of TRPA1 in Skin Physiology and Pathology

**DOI:** 10.3390/ijms22063065

**Published:** 2021-03-17

**Authors:** Roberto Maglie, Daniel Souza Monteiro de Araujo, Emiliano Antiga, Pierangelo Geppetti, Romina Nassini, Francesco De Logu

**Affiliations:** 1Department of Health Sciences, Section of Dermatology, University of Florence, 50139 Florence, Italy; roberto.maglie@unifi.it (R.M.); emiliano.antiga@unifi.it (E.A.); 2Department of Health Sciences, Clinical Pharmacology Unit, University of Florence, 50139 Florence, Italy; daniel.souzamonteirodearaujo@unifi.it (D.S.M.d.A.); pierangelo.geppetti@unifi.it (P.G.); francesco.delogu@unifi.it (F.D.L.)

**Keywords:** TRPA1, TRP channel, skin disease, itch, ion channel, dermatophatology

## Abstract

The transient receptor potential ankyrin 1 (TRPA1), a member of the TRP superfamily of channels, acts as ‘polymodal cellular sensor’ on primary sensory neurons where it mediates the peripheral and central processing of pain, itch, and thermal sensation. However, the TRPA1 expression extends far beyond the sensory nerves. In recent years, much attention has been paid to its expression and function in non-neuronal cell types including skin cells, such as keratinocytes, melanocytes, mast cells, dendritic cells, and endothelial cells. TRPA1 seems critically involved in a series of physiological skin functions, including formation and maintenance of physico-chemical skin barriers, skin cells, and tissue growth and differentiation. TRPA1 appears to be implicated in mechanistic processes in various immunological inflammatory diseases and cancers of the skin, such as atopic and allergic contact dermatitis, psoriasis, bullous pemphigoid, cutaneous T-cell lymphoma, and melanoma. Here, we report recent findings on the implication of TRPA1 in skin physiology and pathophysiology. The potential use of TRPA1 antagonists in the treatment of inflammatory and immunological skin disorders will be also addressed.

## 1. Introduction

Transient receptor potential (TRP) channels are polymodal cation channels primarily permeable to calcium, which work as cellular sensors implicated in many physiological functions, ranging from pure sensory activities, such as nociception and temperature sensation, and homeostatic functions, such as osmoregulation, to many other functions, such as muscle contraction and vasomotor control [1]. In mammals, the superfamily of TRP channels encompasses 28 members [1], behaving as non-selective cation permeable channels, and classified into six subfamilies: The canonical or classic (TRPC1-7), vanilloid (TRPV1-6), melastatin (TRPM1-8), long TRP ankyrin (a solitary member is the transmembrane protein 1 [TRPA1]), and the more distant relatives, polycystins (TRPP1-5) and mucolipins (TRPML1-3) [1,2,3]. TRPs are expressed in a wide variety of both excitable and non-excitable cells [4,5,6,7,8,9,10]. Most TRPs have been localized to the plasma membrane where they non-selectively allow the influx of extracellular cations [11]. However, their presence has been documented in cellular organelles, with a pivotal role in establishing/maintaining vesicular calcium homeostasis and in regulating membrane trafficking [12]. TRPs are considered unique polymodal cell sensors, as their gating is directly operated by a plethora of exogenous and endogenous physical stimuli and chemical mediators or by changes in the intracellular environment [13]. As several TRPs localized in a subset of primary sensory neurons, they result as highly implicated in sensing physiological and noxious agents and more generally in nociceptive stimuli perception in a variety of tissues and organs, including the skin [14,15].

Emerging evidence suggests that multiple TRPs are involved in the regulation of the cutaneous functions. Apart from their prominent expression and role in nociceptive neurons, where they mediate the peripheral and central processing of pain, itch, and thermal sensation [16,17,18,19,20], some TRPs are found in non-neuronal cells [4,5,7] including skin cells [6,21,22,23], where they are critically involved in formation and maintenance of physico-chemical skin barriers, skin cells, and organ growth and differentiation, and cutaneous immunological and inflammatory processes. This review focuses on the functional role of TRPA1 in various cutaneous functions both under physiological and pathophysiological conditions.

## 2. TRPA1 in Skin Physiology

### 2.1. Cutaneous Nerve Fibers and Neurogenic Inflammation

The skin-localized sensory afferents are involved in the neuronal processing of multiple sensory modalities. Aβ-fibers with thickly myelinated axons, thus fast conduction velocities and low activation thresholds, are the predominant class of fibers responsible for sensing light touch. A subpopulation of C-fibers is responsible for gentle touch and light forces similar to Aβ-fibers [24,25,26]. Similar low-threshold Aδ-fiber responses have been observed in humans, but it remains to be determined if these fibers also influence touch perception [27]. The perception of acute noxious or painful touch are typically derived from the activation of high-threshold unmyelinated C-fibers and thinly myelinated Aδ-fibers.

A specific subset of C-fiber and Aδ- fiber nociceptors is exclusively sensitive to the selective TRPV1 agonist, capsaicin, the pungent ingredient in hot chili peppers, and thereby defined as ‘capsaicin-sensitive’ sensory neurons. TRPV1-expressing neurons comprise a subgroup of neurons defined as peptidergic because of their ability to produce neuropeptides, including the calcitonin gene-related peptide (CGRP) and tachykinins, such as substance P (SP) and neurokinin A (NKA) [28,29], which upon peripheral release, cause inflammatory responses, collectively referred to as “neurogenic inflammation” [28,29]. TRPA1 is present in 30–50% of TRPV1-expressing neurons and rarely exists in neurons which do not express TRPV1 [30]. 

The most prominent feature of the TRPA1 resides in its unique sensitivity for several exogenous and endogenous agonists which, based on their structure, activate the channel covalently, or modulate its activity in a different way. A number of naturally occurring TRPA1 agonists mainly found in alimentary sources include herbs and spices, such as cinnamaldehyde, contained in the cinnamon oil extracted from the Cinnamomum [31], several isothiocyanate compounds, such as allyl or benzyl isothiocyanate contained in mustard oil or wasabi, obtained from the Brassica seeds [32], and allicin and diallyl disulfide, contained in garlic (*Allium sativum*) [33]. Other rather heterogenous substances qualified as TRPA1 channel stimulants include volatile irritants, such as acrolein and crotonaldehyde [34,35], chemicals of industrial origin, [36,37,38], general anesthetics (e.g., isoflurane [39], lidocaine [40], propofol [41]), and laboratory chemicals (e.g., formalin [42,43,44]). Additional aldehydes which stimulate TRPA1 are formaldehyde [45], acetaldehyde [46] and crotonaldehyde [35,47] (all contained in cigarette smoke). These compounds, share a reactive chemical structure which enables them to covalently modify specific cysteine residues located within the cytoplasmic N-terminal region of the protein [48], resulting in TRPA1 activation. These features justify the large use of these compounds to better understand the mechanism of action and the role of the channel.

Non-reactive compounds which are unable to modify the channel covalently include compounds from plant origin, such as menthol [49], thymol, and carvacrol [50,51]. The non-electrophilic component contained in Cannabis sativa, delta-9-tetrahydrocannabinol (THC), also activates the TRPA1 channel without producing any covalent modification [52]. Different medicines or their metabolites such as clotrimazole [53], nifedipine [54], and non-steroidal anti-inflammatory drugs, such as diclofenac [55] and acyl-glucuronide ibuprofen [56], represent an additional subgroup of exogenous TRPA1 activators. 

The last ten years have witnessed a series of discoveries that have placed the TRPA1 channel as a major sensor of oxidative, nitrative, and carbonylic stress for the peripheral nervous system. Reactive oxygen (ROS), nitrative (RNS), and carbonylic (RCS) species have shown the ability to gate TRPA1 on peripheral terminals primary sensory neurons, thereby signaling pain, and neurogenic inflammation. ROS activate TRPA1 through a cysteine oxidation or disulfide formation [57], whereas RNS activate the channel through a S-nitrosylation reaction [57].

Among ROS, TRPA1 activators comprise hydrogen peroxide (H_2_O_2_) [37,39,58], hypochlorite (OCl^-^), superoxide (O^2-^) [38]. Among RNS, NO [39,59,60], and peroxynitrite [39] are TRPA1 agonists. Metabolites generated by peroxidation or nitrosylation of plasma membrane phospholipids, including 4-hydroxynonenal (4-HNE), 4-hydroxyhexenal (4-HHE), 4-oxo-2-nonenal (4-ONE), and nitrooleic acid (9-OA-NO_2_) activate TRPA1 channels [37,48,61,62,63]. During inflammation cyclooxygenase induction and activation result in the release of proinflammatory and proalgesic prostaglandings and isoprostanes, which via a non-enzymatic dehydration generate cyclopentenone prostaglandin and isoprostane including 15-deoxy-Δ12,14-PGJ_2_ (15-d-PGJ_2_), PGA_2_ and PGA_1_, and 8-isoprostane-PGA_2_ are formed. Cyclopentenone PGs and iso-PGs have been recognized as TRPA1 activators [64,65]. Finally, among the endogenously produced mediators, the malodourous gas hydrogen sulfide (H_2_S), produced by cysteine metabolism and endowed with vasodilatatory and other properties [66], has also been identified as a TRPA1 stimulant [67].

### 2.2. Keratinocytes Differentiation, Proliferation and Barrier Function

Growing evidence has revealed that TRPs are actively involved in the regulation of skin physiology [68,69,70,71,72,73,74,75,76,77]. TRPV1 expression has been identified in epidermal and hair follicle keratinocytes, dermal mast cells, sebaceous gland-derived sebocytes, and dendritic cells [78,79], which suggest functional roles in homeostatic and ‘sensory’ functions not limited to cutaneous nerve fibers. TRPV2 has been found in keratinocytes [80] and macrophages [81], and TRPV3 in blood vessels [7] and keratinocytes [82]. The presence of TRPV4 has been reported in basal and suprabasal keratinocytes of healthy human skin [22,73]. Finally, TRPA1 has been found in keratinocytes, melanocytes, and fibroblasts [6,23] (Figure 1).

In human skin, immunoreactivity for the TRPA1 channel has been detected in both keratinocytes and melanocytes [6]. It was also observed that the treatment of keratinocytes with icilin, a selective TRPA1 agonist, increased the expression of genes involved in cellular adhesion and extracellular matrix protein synthesis [6].

The production of the stratum corneum is one of the main roles of epidermal keratinocytes. Some studies showed that the administration of TRPA1 activators as well as the application of cold stimuli to the skin of mice, in which the epidermal barrier was mechanically disrupted, accelerated the rate of barrier regeneration [83]. Conversely, the application of a TRPA1 antagonist prevented the beneficial effects and markedly delayed the barrier healing [83]. Moreover, cold-induced TRPA1 activation resulted in a specific increase in intracellular calcium in human cultured epidermal keratinocytes, much higher than that observed in dorsal root ganglion cells [84], thus revealing that epidermis might be more sensitive to low temperature than the peripheral nervous system, and TRPA1 expressed in keratinocytes may have a central role in thermo-sensation of the skin [85]. Despite at low levels, TRPA1 mRNA was recently detected in mouse keratinocytes, where its selective deletion caused a marked deficit in mechanically-evoked ATP release, highlighting a possible involvement of keratinocytes in mechano-transduction [86]. Collectively, these findings suggest a “constitutively active” role for TRPA1 in the epidermal barrier homeostasis. TRPA1 is activated by ultraviolet radiation (UVR) in melanocytes, where its activation by UVR non-detrimental doses results in an early melanin synthesis [23]. However, additional TRPs may contribute to the formation and maintenance of the skin barrier [87], participate in the differentiation and growth of the skin cells [88], and ensure immunological properties during inflammatory processes [88], as TRPV4 activation has been involved in cell survival mechanisms after skin exposure to noxious heat.

## 3. TRPA1 in Skin Diseases

### 3.1. Atopic Dermatitis and Allergic Contact Dermatitis

Atopic dermatitis (AD) and allergic contact dermatitis (ACD) are common inflammatory skin diseases characterized by skin barrier disruption and an inflammatory response dominated by T helper 2 (Th2) cells and related products, such as interleukin (IL)-4, IL-5, and IL-13 [89,90]. Pruritus, which is characteristically histamine-independent, represents the most troublesome symptom of both diseases, resulting in a significantly impaired patient’s quality of life.

TRPA1 contributes to the pathogenesis of chronic [91] and acute histamine-independent pruritus, such as those evoked in mice by injection of chloroquine [92] and the proenkephalin product, BAM8-22 [92,93,94]. In either human or murine AD models, TRPA1 has been shown to be significantly over-expressed by several cell types, including keratinocytes, mast cells, and dermal sensory nerve afferents [91]. TRPA1 expression was also enhanced in cell bodies of dorsal root ganglion (DRG) neurons from AD-mice [91]. In comparison, animal models of ACD revealed an over-expression of TRPA1 only on DRG neuronal cells, while no increased channel expression has been observed in non-neuronal skin cells [95].

A TRPA1-dependent pathway of itch in AD has been firstly identified by using a murine model of the disease induced by IL-13 [91]. In this study, IL-13 caused a chronic AD disease in mice characterized by an intensive chronic itch and increased expression of TRPA1 in mast cells, dermal sensory nerve fibers, and cell bodies of DRG neurons. Interestingly, mast cells recruited by IL-13 and localized in close proximity with TRPA1+ dermal afferents promoted a TRPA1-mediated local secretion of neuropeptides. In addition, pharmacological TRPA1 blockade selectively attenuated the itch-evoked scratching. Genetic deletion of mast cells in these mice led to significant reduction in the itch-scratching behaviors and lowered the TRPA1 expression in dermal neuropeptide containing afferent fibers [91]. Altogether, these data reveal a complex interaction among TRPA1+ dermal afferent nerves and TRPA1+ mast cells in the Th2-mediated inflammatory milieu underlying chronic itch in AD [91] (Figure 1).

An additional study revealed that in a different murine model of AD induced by 2,4-dinitrochlorobenzene (DNCB), genetic deletion of TRPA1 attenuated pathological findings of AD, including ear thickness, epidermal hyperplasia and pruritus, and dermal infiltration by mast cells, Th2 cytokines, and macrophages [96]. Moreover, DNCB, capable of inducing ACD in exposed humans, has been shown to directly and dose-dependently activate TRPA1 [97]. Likewise, in a murine model of ACD induced by topical application of oxazolone [95], TRPA1 deficiency correlated with milder ACD symptoms including pruritus, and lower levels of inflammatory cytokines and T-cell activation. More intriguingly, oxazolone has been shown to directly activate TRPA1, resulting in enhanced release of mediators of neurogenic inflammation and pruritus, including 5-hydroxytryptamine (5-HT), SP, and NKA. In addition, the absence of TRPA1 reduced the number of SP-responsive neurons, which are involved in the central transmission of pain and itch sensations. Similar results were obtained using a mouse model of ACD induced by urushiol, the poison ivy allergen [95]. Recently, in an oxazolone-induced murine model of ACD, the topical application of tacrolimus induced a persistent up-regulation of TRPA1 in DRG neurons and contributed to development of itch, thus explaining the pruritus and stinging sensation produced by the drug in humans [98]. Induction of skin dryness in mice has been associated with TRPA1 activation, that also correlated with changes in the gene expression profile driving to skin hyperplasia and lichenification [99].

In chronic allergic itch, multiple pathways of TRPA1 activation have been shown. These include a keratinocyte-neuron axis based on the release of thymic stromal lymphopoietin [100] and periostin [101], two AD-associated cytokines directly released by keratinocytes, and a Th2-cell-neuronal pathway based on the release of the pruritogenic cytokine IL-31 [102]. Although these studies suggest a role of TRPA and many channel ligands have been studied, yet TRPA1 blockers remain to be used clinically as anti-itch drugs. Off-target effects are a risk for TRPA1 ligands, given TRPA1 broad expression in different neuronal and non-neuronal cellular districts, which encompasses different biological functions. For this reason, the development of such drugs should proceed with caution. Nevertheless, since evidence showed that the TRPA1 is crucially involved in the pathogenesis of AD and ACD, the pharmacological inhibition of the channel could be a valuable complementary strategy for local control of skin inflammation and pruritus observed in both diseases.

### 3.2. Psoriasis

Psoriasis is a common chronic inflammatory skin disease, characterized by erythema, skin thickness, and scaling [103]. Pruritus is observed in 60-90% of the patients [104]. Emerging evidence has highlighted a contribution of nociceptive sensory nerve endings in the pathogenesis of psoriasis, with a multi-faced role in detecting noxious stimuli, promoting the activation of immune cells and modulating the immune microenvironment [105]. Of note, different studies showed increased C-fiber innervation in the epidermis of psoriatic skin lesions [106,107,108,109,110]. The altered quantity of nerve fibers was associated with the increased expression of neuropeptides including SP and CGRP in psoriasis epidermal tissue [108,111]. Elevated neuropeptide content in the plasma of patients with psoriasis also correlated with psoriasis severity index scores [112]. Moreover, cutaneous denervation induces a reduction of skin inflammation in psoriasis patients and in mice with psoriasiform dermatitis [113].

More recently, the role of TRPA1 in psoriasis has been explored. It was originally reported that in a murine model of imiquimod (IMQ)-induced psoriasis-like lesions, topical application of the drug was associated with elevated expression of TRPA1 in affected skin areas [114]. Similar results were obtained in psoriatic skin from human subjects where TRPA1 and TRPV1 genes were over-expressed [115]. Mechanistic studies provided contrasting results on the role of TRPA1 in murine models of psoriasis. Pharmacological blockade or genetic deletion of TRPA1 could, in fact, worsen psoriasis dermatitis and nocifensive and itch behavior in mice, thus suggesting a protective role for TRPA1 in psoriasis [116]. At the molecular level, the pathology in TRPA1 deleted mice was characterized by higher levels of inflammatory cytokines, including IL-1β, TNF-α, and IL-22 compared to wild-type mice [116]. The protective role for TRPA1 in psoriasis was strengthened by the observation that a 3-weeks treatment with dry food containing mustard seed (5%) reduced IMQ inflammation in mice [117]. Conversely, another study reported that TRPA1 genetic deletion sustained the dermal inflammation and the Th17-related cytokines expression in a severe model of IMQ-induced psoriasis, which also induced a systemic inflammatory reaction in mice [114]. As TRPA1 is expressed by primary sensory neurons, keratinocytes, and immune cells, we can speculate that channel function is affected by the immune environment. Collectively, these data suggested that TRPA1 activation or inhibition may simultaneously act in a protective manner in psoriasis, arguably by regulation of the activity of TRPV1 (Figure 1).

### 3.3. Cutaneous T-Cell Lymphoma

Cutaneous T-cell lymphomas (CTCL) are a heterogeneous group of primary cutaneous lympho-proliferative disorders, including mycosis fungoides (MF) and Sézary syndrome (SS) as the most common clinical presentations [118]. Pruritus is a debilitating symptom in patients with MF and SS [118]. In these patients, the shift to Th2-type immunity, with neoplastic cells producing enhanced Th2-associated cytokines, including IL-4 and IL-31, partly explains the severity of pruritus [119,120]. However, a recent study reported TRPA1 as a critical mediator involved in CTCL-associated itch [121]. Recent data showed that miR-711, released by neoplastic skin-resident T-cells, induced TRPA1-dependent itch in mice by direct TRPA1 activation [121]. Interestingly, the itching activity of miR-711 was mediated by a short and evolutionary conserved core sequence that was both necessary and sufficient for the TRPA1 gating. In addition, in contrast with conventional TRPA1 agonists, miR-711 did not homogeneously elicit pain, itch, and neurogenic inflammation [121]. Such diversity was ascribed to a different binding domain in the extracellular portion of the protein, a shorter opening time of TRPA1, and the ensuing lower calcium permeability or the activation of different nerve terminals afferents, which encode diverse sensory modalities. Finally, inhibition of miR-711 activity with an extracellular complementary sequence or disruption of the miR-711/TRPA1 interaction with a blocking peptide significantly attenuated the scratching behavior in a mouse model of CTCL. Collectively, these data revealed an unconventional role of extracellular miRNAs as itch mediators and TRPA1 modulators and confirmed the biological relevance of this interaction in the pathophysiology of CTCL-associated itch.

### 3.4. Other Pruritic Skin or Systemic Diseases

Chronic histamine-independent pruritus occurs in a wide variety of other cutaneous and systemic diseases. A common skin disease characterized by pruritus is scabies, a contagious parasitic infestation caused by the mite Sarcoptes scabiei hominis [122]. In the skin of patients with scabies, non-histaminergic itching receptors, including TRPA1, TRPV1, and the protease-activated receptor 2 (PAR2), have been found to be over-expressed [123]. Interestingly, increased PAR2 expression was associated with increased tryptase+ cells and reduced histamine+ cells near the dermal-epidermal junction, potentially suggesting a TRPA1/mast cell pathway similar to that previously reported in AD [123] (Figure 1).

A common systemic cause of itch is that related to liver dysfunction, which in turn causes an elevation of circulating bile acids (BA). Remarkably, one study reported the co-expression of the G-protein-coupled BA receptor 1 (TGR5) and TRPA1 in cutaneous afferent neurons in mice [124]. TRG5 activation by BA sensitized TRPA1 via enhanced intracellular signaling through Gβγ, protein kinase C, and calcium in vitro. In mice over-expressing TRG5, the exacerbated spontaneous scratching behavior was reduced by TRPA1 antagonists, thus supporting a coactivation of TGR5 and TRPA1 in BA induced pruritus [124].

A study reported TRPA1 over-expression in the epidermis of patients with bullous pemphigoid, a rare autoantibody-mediated blistering disease characterized by intense pruritus, compared to healthy skin, but channel expression did not significantly correlate, neither with eosinophil dermal infiltration nor with the severity of pruritus [125] (Figure 1).

Finally, a distinctive cause of pruritus induced by physical factors is that following burn injuries. Recently, one study revealed that mRNA levels of TRPA1 as well as TRPV4 were increased in the skin of itching burn scars [126]. However, further studies are needed to identify the role of TRPA1 in burn-associated itch.

## 4. Therapeutic Perspectives and Future Directions

The hypothesis that TRPA1 is implicated in chronic neuropathic [9,10,127,128,129], inflammatory [129,130], migraine [131,132], and cancer pain [133] is robustly supported. In contrast, current understanding of the pathophysiological roles of TRPA1 in the skin needs further investigations.

Here we showed that TRPA1 may exert a number of functions in different physiological and pathological skin processes. Increasing evidence indicates a major role of TRPA1 in histamine-independent itch occurring in chronic inflammatory skin diseases such as AD, liver dysfunction, or neoplastic diseases, such as cutaneous T-cell lymphoma, thus making these conditions suitable areas of investigation for drugs that target TRPA1. Among the various TRPA1 antagonists that have shown selectivity toward TRPA1, only five have been tested in clinical trials for the treatment of pain or other conditions (e.g., allergic asthma) [134,135,136,137,138].

A recent review identified 28 patent applications for TRPA1 antagonists from 2015 to 2019. Among them, some have been successfully tested in pre-clinical models of skin diseases such atopic dermatitis [139]. In contrast, none have entered large clinical trials for the treatment of skin diseases in humans. A thorough review of registered clinical trials (www.clinicaltrial.gov) using the key words “TRPA1 and skin” or “TRPA1 and pruritus” finds only one study on the effects of L-menthol, a substance that does not directly target TRPA1, as a topical counter-irritant on cutaneous pain and hyperalgesia provoked by topical application of the TRPA1-agonist trans-cinnamaldehyde in healthy human volunteers. Despite robust evidence of TRPA1 involvement in pruritic skin diseases, reasons explaining the delay in translating TRPA1 antagonists from the pre-clinical to the clinical setting, remain unclear. However, other therapeutic strategies are worth exploring.

It is noteworthy that multiple molecules can selectively activate TRPA1 inducing long-lasting desensitization of the channel. Crotalphine has been shown to inhibit a chemically induced inflammatory hypersensitivity in mice via the desensitization of TRPA1-peptidergic nerve endings [140]. In allergic rhinitis, a disease sharing some pathological features with AD and ACD, a combination of azelastine hydrochloride and fluticasone propionate has been shown to induce desensitization of TRPA1 and TRPV1 expressing sensory neurons, thus ameliorating local airway inflammation [141]. Finally, after the exposure to isopetasin, extracts from the butterbur plant (*Petasites hybridus*), and parthenolide, a major constituent of *Tanacetum parthenium*, TRPA1 channel and the TRPA1 expressing neurons undergo to a dose-dependent neuronal desensitization which may account for the relief of pain and neurogenic inflammation of the two plant extracts [142,143].

TRPA1 remains a field of active investigation for other skin conditions. For example, TRPA1 proteins have been found in melanoma cell lines [144,145,146], but the role of TRPA1 in melanoma in vivo remains far less clear. TRPA1 is of paramount importance to signal pain associated with skin cancers or related therapies, including dacarbazine-induced pain in melanoma [147], or pain associated with photodynamic therapy used for the treatment of non-melanoma skin cancer [148]. Likewise, it may be intriguing to explore TRPA1 activity in pain associated with inflammatory or autoimmune skin diseases, such as pemphigus vulgaris [149].

Finally, increasing evidence supports a role for TRPA1 in fibrosis associated with systemic diseases [150,151]. Accordingly, the possibility that TRPA1 promotes fibrosis in fibrogenic skin diseases (e.g., scleroderma) or wound healing is worth investigating, due to the lack of effective therapeutic strategies in such conditions.

## Figures and Tables

**Figure 1 ijms-22-03065-f001:**
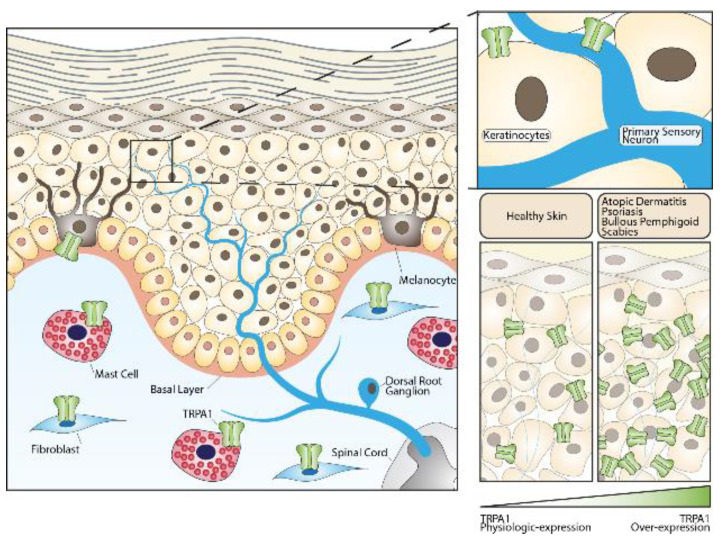
Role of the TRPA1 channel in skin homeostasis and skin diseases. TRPA1 channels expressed in sensory fibers innervating the skin or in different non-neuronal cells can either contribute to maintaining normal skin physiology or play important roles in the pathogenesis of skin diseases.

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
