# Peer review of "The Role of TRPA1 in Skin Physiology and Pathology"

_ijms, 2021, doi:10.3390/ijms22063065_

Round 1

Reviewer 1 Report

The review is well structured and written.  I have some minor concerns.

  1. References 112-113 are unavailable. Does Hydra Biosciences still exist? It is better to find other references. 
  2.  Chapter 4. Therapeutic perspectives and future directions very poorly describe the perspectives and directions. As a matter of fact, low molecular weight antagonists of TRPA1 (and TRPV1) failed in preclinical and clinical trials. But other opportunities are still open - weak activation/potentiation that leads to desensitization of channel (or cell) to external stimuli. The authors can improve this chapter. 

Author Response

Reviewer 1

The review is well structured and written.  I have some minor concerns.

References 112-113 are unavailable. Does Hydra Biosciences still exist? It is better to find other references.

Chapter 4. Therapeutic perspectives and future directions very poorly describe the perspectives and directions. As a matter of fact, low molecular weight antagonists of TRPA1 (and TRPV1) failed in preclinical and clinical trials. But other opportunities are still open - weak activation/potentiation that leads to desensitization of channel (or cell) to external stimuli. The authors can improve this chapter. 

We thank the reviewer for the comments. Following reviewer suggestion, the last chapter has been carefully revised (Line 319).

Reviewer 2 Report

The manuscript ijms-1151922, named “The role of TRPA1 in skin physiology and pathology’ represents a short review on the TRPA1 channel and its particular role in the skin. There are some corrections needed to improve the paper.

The authors should use the journal’s template. Check the mdpi writing style for all the figures and tables and correct it.

The section R78-R100 describes mainly other TRP channels. It should be focused more on TRPA1. The authors should eliminate all unrelated data and present substances that interact with TRPA1 receptors. See the article:

TRPA1: a molecular view, J Neurophysiol. 2019 Feb 1;121(2):427-443

and the authors’ own article TRPA1 as a therapeutic target for nociceptive pain

row 94, mustard oil is not exactly the same as allyl isothiocyanate (AITC). Avoid any confusion for the readers.

Row 192, detail on pharmacological tools that can attenuate the itch-evoked scratching by targeting TRPA1. The review must be more that a collection of data. The authors should comment on the future perspective of such drugs to be used in the future. Are there possible downsides to this approach?

Row 219, present the doses of AITC and how was administered.

In section 4, Therapeutic perspectives and future directions, give more details on drugs under clinical studies, drugs targeting TRPA1, even if not directed at skin conditions. The discussion on this section should be developed in order to demonstrate that TRPA1 is a viable therapeutical target for skin conditions. In my personal opinion, the available data is to limited and not very convincing. The authors should present and argument their own opinion.

Author Response

Reviewer 2

The manuscript ijms-1151922, named “The role of TRPA1 in skin physiology and pathology’ represents a short review on the TRPA1 channel and its particular role in the skin. There are some corrections needed to improve the paper.

The authors should use the journal’s template. Check the mdpi writing style for all the figures and tables and correct it.

We thank the reviewer for the comments. The revised manuscript has been formatted following journal’s template and style.

The section R78-R100 describes mainly other TRP channels. It should be focused more on TRPA1. The authors should eliminate all unrelated data and present substances that interact with TRPA1 receptors. See the article:

TRPA1: a molecular view, J Neurophysiol. 2019 Feb 1;121(2):427-443

and the authors’ own article TRPA1 as a therapeutic target for nociceptive pain

Thanks for this comment. The section now focuses on TRPA1 channel by reporting a detailed description of different activating/modulating substances (Line 67).

row 94, mustard oil is not exactly the same as allyl isothiocyanate (AITC). Avoid any confusion for the readers.

Thanks to the reviewer for the additional comment. The section has been changed (Line 67).

Row 192, detail on pharmacological tools that can attenuate the itch-evoked scratching by targeting TRPA1. The review must be more that a collection of data. The authors should comment on the future perspective of such drugs to be used in the future. Are there possible downsides to this approach?

Following reviewer’s suggestion, a comment has been added in this section (Line 213).

Row 219, present the doses of AITC and how was administered.

The information on dose and time of treatment described in the cited article have been added in the revised version (Line 246).

In section 4, Therapeutic perspectives and future directions, give more details on drugs under clinical studies, drugs targeting TRPA1, even if not directed at skin conditions. The discussion on this section should be developed in order to demonstrate that TRPA1 is a viable therapeutical target for skin conditions. In my personal opinion, the available data is to limited and not very convincing. The authors should present and argument their own opinion.

Following reviewer’s comment, the last paragraph has been implemented (Line 319).